# AtlasODE: Learning Continuous Atlases via Neural Ordinary Differential Equations

**Jichang Zhang** [1] ZHANGJICHANG@MAIL.BNU.EDU.CN

**Feng Li** [1] FengLi@MAIL.BNU.EDU.CN

**Tongtong Che** [1] CHE@BNU.EDU.CN

**Shuyu Li** [*1] SHUYULI@BNU.EDU.CN

[1] *State Key Laboratory of Cognitive Neuroscience and Learning, Beijing Normal University*

## Abstract

Current approaches to brain atlas generation rely on independent static fitting, inherently failing to capture continuous morphological evolution. To address this gap, we propose `AtlasODE`, a novel framework leveraging Neural Ordinary Differential Equations to reframe atlas generation as a continuous-time integration process. To explicitly model neurodegeneration, our Pathological Trajectory Modulator (PTM) formulates disease as an adaptive perturbation, characterizing pathological evolution as a divergence from the normative aging trajectory. Experimental results on synthetic toy data and brain MRI data demonstrate the superior performance of `AtlasODE`.

**Keywords:** Anatomical atlas, Brain template, Neural ODE, Aging and disease evolution

## 1. Introduction

Anatomical atlases, also known as templates, serve as essential reference frames for neuroimaging analysis (Alexander-Bloch et al., 2013; Leergaard and Bjaalie, 2022). A single general population atlas inherently fails to capture the complex structural variability across the human lifespan. Therefore, recent studies increasingly advocate for constructing age-conditional atlases. The construction of these atlases traditionally relies on group-wise registration (Avants and Gee, 2004; Che et al., 2025; He and Chung, 2025) or conditional generative models (Dalca et al., 2019; Dey et al., 2021; Starck et al., 2025) which yield static atlases specific to age-subgroups. Since brain development and neurodegeneration are continuous dynamical processes (Bethlehem et al., 2022), relying on isolated group-wise fitting cannot capture these morphological evolutions. To bridge this critical gap, we propose a novel method, termed `AtlasODE`, that leverages Neural Ordinary Differential Equations (NODE (Chen et al., 2018)) to explicitly model the underlying continuous dynamics of brain morphology for learning atlases. Moreover, by modulating the aging dynamics based on cognitive status within the NODE framework, we introduce a biologically interpretable mechanism to learn disease-specific atlases, effectively capturing pathological deviations.

## 2. Methods

**Overview and Learning Process.** `AtlasODE` consists of a NODE-based generator producing continuous atlases $\bar{\mathcal{I}}^c$, and a registration network predicting the deformation field

---

* Corresponding author

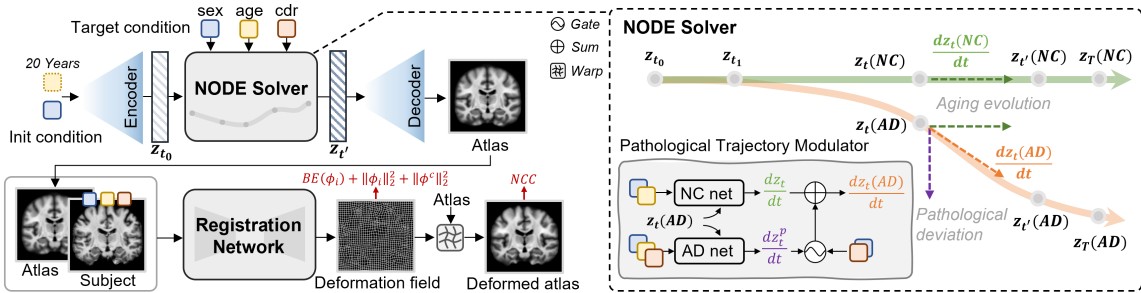

Figure 1: Illustration of the proposed methods of `AtlasODE`.

$\phi_i$ from $\bar{\mathcal{I}}^c$ to $\mathcal{I}_i^c$. The optimal atlas is obtained by minimizing the average morphological distance across the cohort specific to the condition $c$, formulated as:

$$\mathcal{L} = NCC(\bar{\mathcal{I}}^c \circ \phi_i, \mathcal{I}_i^c) + \beta_1 BE(\phi_i) + \beta_2 ||\phi_i||_2^2 + \beta_3 ||\bar{\phi}||_2^2, \tag{1}$$

where $\circ$ denotes spatial transformation, $\bar{\phi}$ the moving average of dataset-wise $\{\phi_i\}$, and NCC/BE the normalized cross-correlation and bending energy. $c$ includes normalized age $t$, sex $c_s$, and disease status $c_p$. The pipeline of `AtlasODE` is illustrated in Figure 1.

**Generator Based on NODE.** We formulate normative morphological evolution via NODE in the latent space. Given an initial latent state $z_{t_0}$ at a baseline age $t_0$ (20 years) and specific sex $c_s$, the state $z_{t'}$ at any target normalized age $t'$ is computed by integrating the aging evolution parameterized by an NC net $f_{NC}$:

$$z_{t'} = z_{t_0} + \int_{t_0}^{t'} f_{NC}(z_t, t, c_s)dt \tag{2}$$

This continuous integration guarantees temporal smoothness and topology preservation, capturing the intrinsic manifold of normative aging.

**Pathological Trajectory Modulator.** To mathematically decouple disease progression from normative aging, we introduce a Pathological Trajectory Modulator (PTM). Instead of treating the disease $c_p$ as an entangled static condition, we formulate it as a perturbation superimposed onto the healthy baseline via an AD net $f_{AD}$ and a gate net $g$:

$$\frac{dz_t}{dt} = f_{NC}(z_t, t, c_s) + c_p \cdot g(c_s, c_p) \cdot f_{AD}(z_t, t, c_s, c_p) \tag{3}$$

The gate $g(c_s, c_p)$ captures pathological deviations by modulating the trajectory's divergence from the aging evolution based on sex and disease status.

## 3. Experiments

**Synthetic Toy Data.** We construct a synthetic toy dataset to validate the smooth and continuous modeling capability of `AtlasODE`. At the condition-level, condition variable $c \in [0.1, 1.0]$ explicitly controls the spatial translation and volumetric scaling. To simulate biological irregularities, we generate 500 instance-level samples per condition using randomized Fourier harmonics, spatial shifts, blur, and noise.

**Brain MRI Data.** We utilize the OASIS dataset (Marcus et al., 2007) to evaluate the clinical efficacy of our framework. The cohort comprises T1-weighted MRI scans of 416 subjects

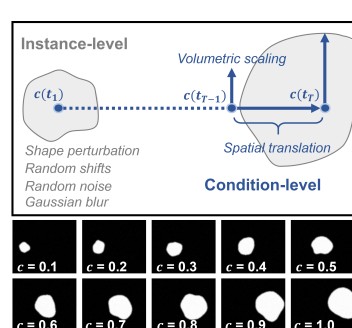

Figure 2: Synthetic data.

with a wide age span from 18 to 96 years (rescaled to a [20, 90] range). Metadata include biological sex and Clinical Dementia Rating (CDR) scores.

**Implementation Details.** For synthetic data, we employ AtlasMorph (Dalca et al., 2019) as the baseline. For the brain MRI dataset, both our `AtlasODE` and the baseline incorporate an adversarial discriminator, following AtlasGAN (Dey et al., 2021). All configurations use convolutional encoders and decoders. The continuous latent trajectory is numerically integrated via a fixed-step Euler solver (step size 0.1), coupled with the adjoint sensitivity method (Chen et al., 2018) to compute gradients at a low memory cost.

## 4. Results and Conclusion

As shown in Figure 3 Top, compared to the baseline, `AtlasODE` exhibits superior capability in maintaining geometric consistency and preserving topological structures during continuous morphological modeling. Additionally, `AtlasODE` achieves a 8.5% improvement in atlas centrality under unseen conditions ($||\bar{\phi}||_2^2$: 194 vs. 212 for baseline). On real brain MRI data (Figure 3 Bottom), `AtlasODE` generates sharp and smoothly varying atlases. Furthermore, the proposed PTM enables the generated disease-specific atlases to progressively diverge from the normative aging trajectory, which aligns with the findings in (Coupé et al., 2019). In conclusion, `AtlasODE` reframes atlas generation as continuous-time integration, capturing the morphological manifold while providing an interpretable mechanism to adaptively model disease evolution.

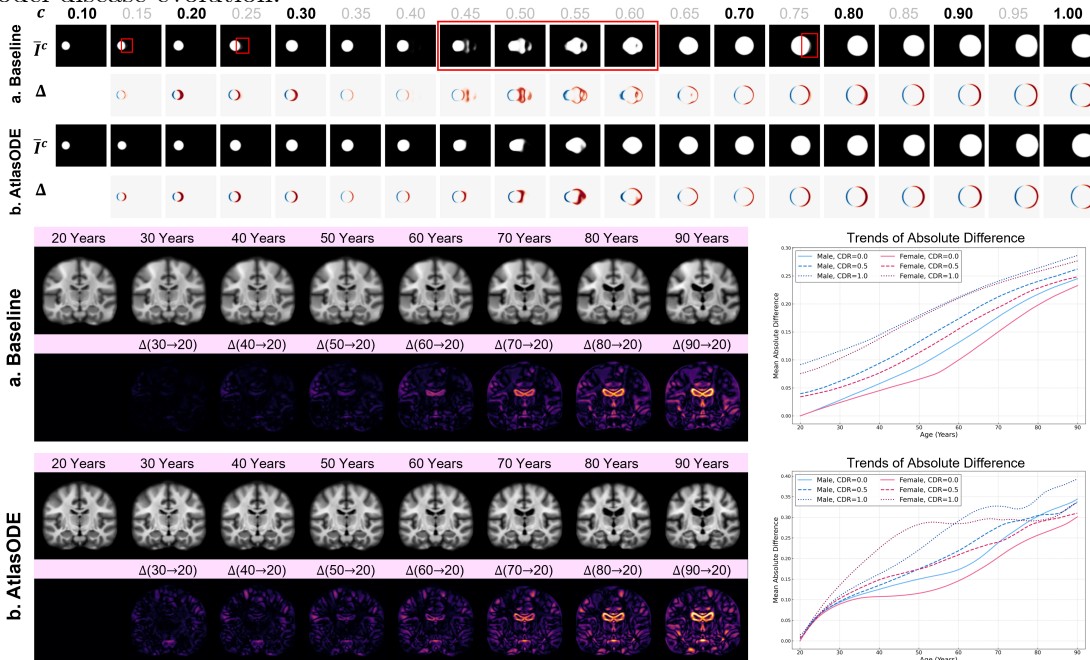

Figure 3: Atlas generation results. **Top**: Synthetic data, with grey $c$ denoting unseen or hypothetical conditions during training and $\Delta$ the adjacent atlas difference. **Bottom**: Brain MRI, with $\Delta$ being the absolute difference from the 20-year atlas.

## Acknowledgments

National Natural Science Foundation of China [Nos. 32271146 and 81622025], and the Startup Funds for Talents at Beijing Normal University, China.

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
