# OpenReview forum: "AtlasODE: Learning Continuous Atlases via Neural Ordinary Differential Equations"
_MIDL.io/2026/Short_Papers — MIDL 2026 - Short Papers Poster_

### Official Review · Reviewer_fmkV · 2026-05-02
**Learning Continuous Atlases via Neural ODEs**

**Rating:** 3
**Confidence:** 4

**Review:**

Key implementation details such as hyperparameters, training schedules, and initialization are not fully specified. The clinical relevance of disease modeling is not demonstrated, as no downstream task or clinical metric is evaluated.
Overall, the work presents an incremental extension of existing generative modeling approaches with insufficient experimental validation.
The formulation of continuous atlas evolution via NODEs closely follows prior work on continuous latent dynamics and generative modeling. Similarly, conditional atlas generation using deep learning and generative models has already been explored extensively, including conditional deformable templates and GAN-based atlas construction. The proposed Pathological Trajectory Modulator is an additive perturbation term.
The paper proposes a continuous atlas generation framework using Neural Ordinary Differential Equations. While the idea of modeling temporal evolution is conceptually appealing, the work does not demonstrate sufficient novelty or experimental rigor.
The synthetic experiment is trivial and does not reflect realistic anatomical variability. The real-data experiment on OASIS lacks quantitative evaluation beyond a single metric (atlas centrality), reported without variance, confidence intervals, or statistical testing. The reported improvement (8.5 percent) is not contextualized and may fall within noise.
The use of a fixed-step Euler solver is simplistic and may introduce numerical instability. There is no analysis of sensitivity to solver parameters or integration error. The model is trained and evaluated on a relatively small dataset (416 subjects), with no cross-validation or robustness analysis.
There is no comparison against strong modern baselines such as diffusion-based atlas generation or recent conditional generative models. The evaluation relies heavily on qualitative visualization.

**Summary:**

This paper proposes AtlasODE, a framework that models atlas generation as a continuous process using Neural Ordinary Differential Equations. The method introduces a trajectory formulation for aging and a modulation mechanism for disease progression. Experiments are conducted on synthetic data and the OASIS dataset, showing qualitative improvements and a reported reduction in atlas centrality error. The work aims to capture continuous morphological evolution rather than discrete atlas construction.

**Strengths:**

The paper addresses an important limitation of static atlas construction by attempting to model continuous morphological changes. The integration of temporal dynamics into atlas generation is conceptually meaningful. The formulation is mathematically clear and the framework is relatively interpretable.

**Weaknesses:**

The methodological novelty is limited, as the approach largely combines existing NODE formulations with standard atlas generation pipelines. The experimental evaluation is weak, relying on qualitative results and a single metric without statistical analysis. There is no comparison to strong recent baselines, and the dataset is small. The claims regarding disease modeling are not supported by quantitative evidence. Reproducibility is limited due to missing implementation details.

**Justification Of Rating:**

The evaluation is statistically weak and does not support the claims. The contribution is incremental and not competitive with current state of the art but there is a hint of contribution so perhaps good enough for a short paper.

---

### Decision · Program_Chairs · 2026-05-08

Accept (Poster)